# On Detecting Adversarial Perturbations

**Jan Hendrik Metzen & Tim Genewein & Volker Fischer & Bastian Bischoff**
Bosch Center for Artificial Intelligence, Robert Bosch GmbH
Robert-Bosch-Campus 1, 71272 Renningen, Germany
`JanHendrik.Metzen@de.bosch.com`

## Abstract

Machine learning and deep learning in particular has advanced tremendously on perceptual tasks in recent years. However, it remains vulnerable against adversarial perturbations of the input that have been crafted specifically to fool the system while being quasi-imperceptible to a human. In this work, we propose to augment deep neural networks with a small "detector" subnetwork which is trained on the binary classification task of distinguishing genuine data from data containing adversarial perturbations. Our method is orthogonal to prior work on addressing adversarial perturbations, which has mostly focused on making the classification network itself more robust. We show empirically that adversarial perturbations can be detected surprisingly well even though they are quasi-imperceptible to humans. Moreover, while the detectors have been trained to detect only a specific adversary, they generalize to similar and weaker adversaries. In addition, we propose an adversarial attack that fools both the classifier and the detector and a novel training procedure for the detector that counteracts this attack.

## 1 Introduction

In the last years, machine learning and in particular deep learning methods have led to impressive performance on various challenging perceptual tasks, such as image classification (Russakovsky et al., 2015; He et al., 2016) and speech recognition (Amodei et al., 2016). Despite these advances, perceptual systems of humans and machines still differ significantly. As Szegedy et al. (2014) have shown, small but carefully directed perturbations of images can lead to incorrect classification with high confidence on artificial systems. Yet, for humans these perturbations are often visually imperceptible and do not stir any doubt about the correct classification. In fact, so called *adversarial examples* are crucially characterized by requiring minimal perturbations that are quasi-imperceptible to a human observer. For computer vision tasks, multiple techniques to create such adversarial examples have been developed recently. Perhaps most strikingly, adversarial examples have been shown to transfer between different network architectures, and networks trained on disjoint subsets of data (Szegedy et al., 2014). Adversarial examples have also been shown to translate to the real world (Kurakin et al., 2016), e.g., adversarial images can remain adversarial even after being printed and recaptured with a cell phone camera. Moreover, Papernot et al. (2016a) have shown that a potential attacker can construct adversarial examples for a network of unknown architecture by training an auxiliary network on similar data and exploiting the transferability of adversarial inputs.

The vulnerability to adversarial inputs can be problematic and even prevent the application of deep learning methods in safety- and security-critical applications. The problem is particularly severe when human safety is involved, for example in the case of perceptual tasks for autonomous driving. Methods to increase robustness against adversarial attacks have been proposed and range from augmenting the training data (Goodfellow et al., 2015) over applying JPEG compression to the input (Dziugaite et al., 2016) to distilling a hardened network from the original classifier network (Papernot et al., 2016b). However, for some recently published attacks (Carlini & Wagner, 2016), no effective counter-measures are known yet.

In this paper, we propose to train a binary detector network, which obtains inputs from intermediate feature representations of a classifier, to discriminate between samples from the original data set and adversarial examples. Being able to detect adversarial perturbations might help in safety- and security-critical semi-autonomous systems as it would allow disabling autonomous operation and

requesting human intervention (along with a warning that someone might be manipulating the system). However, it might intuitively seem very difficult to train such a detector since adversarial inputs are generated by tiny, sometimes visually imperceptible, perturbations of genuine examples. Despite this intuition, our results on CIFAR10 and a 10-class subset of ImageNet show that a detector network that achieves high accuracy in detection of adversarial inputs can be trained successfully. Moreover, while we train a detector network to detect perturbations of a specific adversary, our experiments show that detectors generalize to similar and weaker adversaries. An obvious attack against our approach would be to develop adversaries that take into account both networks, the classification and the adversarial detection network. We present one such adversary and show that we can harden the detector against such an adversary using a novel training procedure.

## 2 BACKGROUND

Since their discovery by Szegedy et al. (2014), several methods to generate adversarial examples have been proposed. Most of these methods generate adversarial examples by optimizing an image w.r.t. the linearized classification cost function of the classification network by maximizing the probability for all but the true class or minimizing the probability of the true class (e.g., (Goodfellow et al., 2015), (Kurakin et al., 2016)). The method introduced by Moosavi-Dezfooli et al. (2016b) estimates a linearization of decision boundaries between classes in image space and iteratively shifts an image towards the closest of these linearized boundaries. For more details about these methods, please refer to Section 3.1.

Several approaches exist to increase a model's robustness against adversarial attacks. Goodfellow et al. (2015) propose to augment the training set with adversarial examples. At training time, they minimize the loss for real and adversarial examples, while adversarial examples are chosen to fool the current version of the model. In contrast, Zheng et al. (2016) propose to append a stability term to the objective function, which forces the model to have similar outputs for samples of the training set and their perturbed versions. This differs from data augmentation since it encourages smoothness of the model output between original and distorted samples instead of minimizing the original objective on the adversarial examples directly. Another defense-measure against certain adversarial attack methods is defensive distillation (Papernot et al., 2016b), a special form of network distillation, to train a network that becomes almost completely resistant against attacks such as the L-BFGS attack (Szegedy et al., 2014) and the fast gradient sign attack (Goodfellow et al., 2015). However, Carlini & Wagner (2016) recently introduced a novel method for constructing adversarial examples that manages to (very successfully) break many defense methods, including defensive distillation. In fact, the authors find that previous attacks were very fragile and could easily fail to find adversarial examples even when they existed. An experiment on the cross-model adversarial portability (Rozsa et al., 2016) has shown that models with higher accuracies tend to be more robust against adversarial examples, while examples that fool them are more portable to less accurate models.

Even though the existence of adversarial examples has been demonstrated several times on many different classification tasks, the question of why adversarial examples exist in the first place and whether they are sufficiently regular to be detectable, which is studied in this paper, has remained open. Szegedy et al. (2014) speculated that the data-manifold is filled with "pockets" of adversarial inputs that occur with very low probability and thus are almost never observed in the test set. Yet, these pockets are dense and so an adversarial example is found virtually near every test case. The authors further speculated that the high non-linearity of deep networks might be the cause for the existence of these low-probability pockets. Later, Goodfellow et al. (2015) introduced the *linear explanation*: Given an input and some adversarial noise $\eta$ (subject to: $||\eta||_\infty < \epsilon$), the dot product between a weight vector $w$ and an adversarial input $x^{\text{adv}} = x + \eta$ is given by $w^{\text{T}} x^{\text{adv}} = w^{\text{T}} x + w^{\text{T}} \eta$. The adversarial noise $\eta$ causes a neuron's activation to grow by $w^{\text{T}} \eta$. The max-norm constraint on $\eta$ does not allow for large values in one dimension, but if $x$ and thus $\eta$ are high-dimensional, many small changes in each dimension of $\eta$ can accumulate to a large change in a neuron's activation. The conclusion was that "linear behavior in high-dimensional spaces is sufficient to cause adversarial examples".

Tanay & Griffin (2016) challenged the linear-explanation hypothesis by constructing classes of images that do not suffer from adversarial examples under a linear classifier. They also point out that if the change in activation $w^{\text{T}} \eta$ grows linearly with the dimensionality of the problem, so does the activation

$w^{\mathrm{T}}x$. Instead of the linear explanation, Tanay et al. provide a different explanation for the existence of adversarial examples, including a strict condition for the non-existence of adversarial inputs, a novel measure for the strength of adversarial examples and a taxonomy of different classes of adversarial inputs. Their main argument is that if a learned class boundary lies close to the data manifold, but the boundary is (slightly) tilted with respect to the manifold[1], then adversarial examples can be found by perturbing points from the data manifold towards the classification boundary until the perturbed input crosses the boundary. If the boundary is only slightly tilted, the distance required by the perturbation to cross the decision-boundary is very small, leading to strong adversarial examples that are visually almost imperceptibly close to the data. Tanay et. al further argue that such situations are particularly likely to occur along directions of low variance in the data and thus speculate that adversarial examples can be considered an effect of an over-fitting phenomenon that could be alleviated by proper regularization, though it is completely unclear how to regularize neural networks accordingly.

Recently, Moosavi-Dezfooli et al. (2016a) demonstrated that there even exist universal, image-agnostic perturbations which, when added to all data points, fool deep nets on a large fraction of ImageNet validation images. Moreover, they showed that these universal perturbations are to a certain extent also transferable between different network architectures. While this observation raises interesting questions about geometric properties and correlations of different parts of the decision boundary of deep nets, potential regularities in adversarial perturbations may also help detecting them. However, the existence of universal perturbations does not necessarily imply that the adversarial examples generated by data-dependent adversaries will be regular. Actually, Moosavi-Dezfooli et al. (2016a) show that universal perturbations are not unique and that there even exist many different universal perturbations which have little in common. This paper studies if data-dependent adversarial perturbations can nevertheless be detected reliably and answers this question affirmatively.

## 3 METHODS

In this section, we introduce the adversarial attacks used in the experiments, propose an approach for detecting adversarial perturbations, introduce a novel adversary that aims at fooling both the classification network and the detector, and propose a training method for the detector that aims at counteracting this novel adversary.

### 3.1 GENERATING ADVERSARIAL EXAMPLES

Let $x$ be an input image $x \in \mathbb{R}^{3 \times \text{width} \times \text{height}}$, $y_{\text{true}}(x)$ be a one-hot encoding of the true class of image $x$, and $\mathrm{J_{cls}}(x, y(x))$ be the cost function of the classifier (typically cross-entropy). We briefly introduce different adversarial attacks used in the remainder of the paper.

**Fast method:**   One simple approach to compute adversarial examples was described by Goodfellow et al. (2015). The applied perturbation is the direction in image space which yields the highest increase of the linearized cost function under $\ell_\infty$-norm. This can be achieved by performing one step in the direction of the gradient's sign with step-width $\varepsilon$:

$$x^{\text{adv}} = x + \varepsilon \, \mathrm{sgn}(\nabla_x \mathrm{J_{cls}}(x, y_{\text{true}}(x)))$$

Here, $\varepsilon$ is a hyper-parameter governing the distance between adversarial and original image. As suggested in Kurakin et al. (2016) we also refer to this as the *fast method* due to its non-iterative and hence fast computation.

**Basic Iterative method ($\ell_\infty$ and $\ell_2$):**   As an extension, Kurakin et al. (2016) introduced an iterative version of the fast method, by applying it several times with a smaller step size $\alpha$ and clipping all pixels after each iteration to ensure results stay in the $\varepsilon$-neighborhood of the original image:

$$x_0^{\text{adv}} = x, \quad x_{n+1}^{\text{adv}} = \mathrm{Clip}_x^\varepsilon \left\{ x_n^{\text{adv}} + \alpha \, \mathrm{sgn}(\nabla_x \mathrm{J_{cls}}(x_n^{\text{adv}}, y_{\text{true}}(x))) \right\}$$

---

[1]It is easier to imagine a linear decision-boundary - for neural networks this argument must be translated into a non-linear equivalent of boundary tilting.

Following Kurakin et al. (2016), we refer to this method as the *basic iterative method* and use $\alpha = 1$, i.e., we change each pixel maximally by 1. The number of iterations is set to 10. In addition to this method, which is based on the $\ell_\infty$-norm, we propose an analogous method based on the $\ell_2$-norm: in each step this method moves in the direction of the (normalized) gradient and projects the adversarial examples back on the $\varepsilon$-ball around $x$ (points with $\ell_2$ distance $\varepsilon$ to $x$) if the $\ell_2$ distance exceeds $\varepsilon$:

$$x_0^{\mathrm{adv}} = x, \quad x_{n+1}^{\mathrm{adv}} = \mathrm{Project}_x^\varepsilon \left\{ x_n^{\mathrm{adv}} + \alpha \frac{\nabla_x \mathrm{J}_{\mathrm{cls}}(x_n^{\mathrm{adv}}, y_{\mathrm{true}}(x))}{||\nabla_x \mathrm{J}_{\mathrm{cls}}(x_n^{\mathrm{adv}}, y_{\mathrm{true}}(x))||_2} \right\}$$

**DeepFool method:** Moosavi-Dezfooli et al. (2016b) introduced the DeepFool adversary which iteratively perturbs an image $x_0^{\mathrm{adv}}$. Therefore, in each step the classifier is linearized around $x_n^{\mathrm{adv}}$ and the closest class boundary is determined. The minimal step according to the $\ell_p$ distance from $x_n^{\mathrm{adv}}$ to traverse this class boundary is determined and the resulting point is used as $x_{n+1}^{\mathrm{adv}}$. The algorithm stops once $x_{n+1}^{\mathrm{adv}}$ changes the class of the actual (not linearized) classifier. Arbitrary $\ell_p$-norms can be used within DeepFool, and here we focus on the $\ell_2$- and $\ell_\infty$-norm. The technical details can be found in (Moosavi-Dezfooli et al., 2016b). We would like to note that we use the variant of DeepFool presented in the first version of the paper (`https://arxiv.org/abs/1511.04599v1`) since we found it to be more stable compared to the variant reported in the final version.

## 3.2 Detecting Adversarial Examples

We augment classification networks by (relatively small) subnetworks, which branch off the main network at some layer and produce an output $p_{adv} \in [0, 1]$ which is interpreted as the probability of the input being adversarial. We call this subnetwork "adversary detection network" (or "detector" for short) and train it to classify network inputs into being regular examples or examples generated by a specific adversary. For this, we first train the classification networks on the regular (non-adversarial) dataset as usual and subsequently generate adversarial examples for each data point of the train set using one of the methods discussed in Section 3.1. We thus obtain a balanced, binary classification dataset of twice the size of the original dataset consisting of the original data (label zero) and the corresponding adversarial examples (label one). Thereupon, we freeze the weights of the classification network and train the detector such that it minimizes the cross-entropy of $p_{adv}$ and the labels. The details of the adversary detection subnetwork and how it is attached to the classification network are specific for datasets and classification networks. Thus, evaluation and discussion of various design choices of the detector network are provided in the respective section of the experimental results.

## 3.3 Dynamic Adversaries and Detectors

In the worst case, an adversary might not only have access to the classification network and its gradient but also to the adversary detector and its gradient[2]. In this case, the adversary might potentially generate inputs to the network that fool both the classifier (i.e., get classified wrongly) and fool the detector (i.e., look innocuous). In principle, this can be achieved by replacing the cost $\mathrm{J}_{\mathrm{cls}}(x, y_{\mathrm{true}}(x))$ by $(1 - \sigma)\mathrm{J}_{\mathrm{cls}}(x, y_{\mathrm{true}}(x)) + \sigma\mathrm{J}_{\mathrm{det}}(x, 1)$, where $\sigma \in [0, 1]$ is a hyperparameter and $\mathrm{J}_{\mathrm{det}}(x, 1)$ is the cost (cross-entropy) of the detector for the generated $x$ and the label one, i.e., being adversarial. An adversary maximizing this cost would thus aim at letting the classifier mis-label the input $x$ and making the detectors output $p_{adv}$ as small as possible. The parameter $\sigma$ allows trading off these two objectives. For generating $x$, we propose the following extension of the basic iterative ($\ell_\infty$) method:

$$x_0^{\mathrm{adv}} = x; \ x_{n+1}^{\mathrm{adv}} = \mathrm{Clip}_x^\varepsilon \left\{ x_n^{\mathrm{adv}} + \alpha \left[ (1-\sigma)\,\mathrm{sgn}(\nabla_x \mathrm{J}_{\mathrm{cls}}(x_n^{\mathrm{adv}}, y_{\mathrm{true}}(x))) + \sigma\,\mathrm{sgn}(\nabla_x \mathrm{J}_{\mathrm{det}}(x_n^{\mathrm{adv}}, 1)) \right] \right\}$$

Note that we found a smaller $\alpha$ to be essential for this method to work; more specifically, we use $\alpha = 0.25$. Since such an adversary can adapt to the detector, we call it a *dynamic adversary*. To

---

[2]We would like to emphasize that is a stronger assumption than granting the adversary access to only the original classifier's predictions and gradients since the classifier's predictions need often be presented to a user (and thus also to an adversary). The same is typically not true for the predictions of the adversary detector as they will only be used internally.

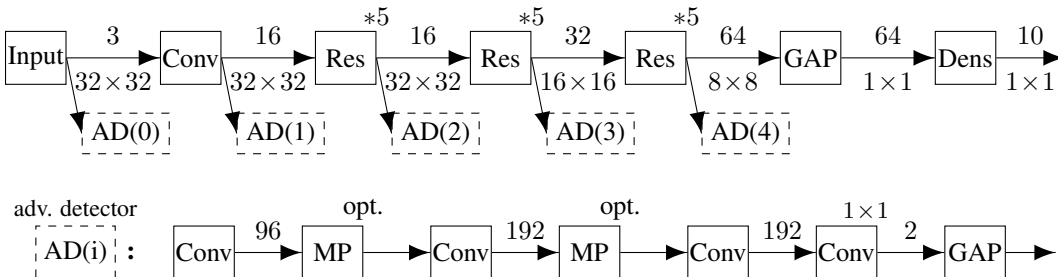

Figure 1: (Top) ResNet used for classification. Numbers on top of arrows denote the number of feature maps and numbers below arrows denote spatial resolutions. `Conv` denotes a convolutional layer, `Res`$^{*5}$ denotes a sequence of 5 residual blocks as introduced by He et al. (2016), `GAP` denotes a global-average pooling layer and `Dens` a fully-connected layer. Spatial resolutions are decreased by strided convolution and the number of feature maps on the residual's shortcut is increased by 1x1 convolutions. All convolutional layers have 3x3 receptive fields and are followed by batch normalization and rectified linear units. (Bottom) Topology of detector network, which is attached to one of the AD(i) positions. `MP` denotes max-pooling and is optional: for AD(3), the second pooling layer is skipped, and for AD(4), both pooling layers are skipped.

counteract dynamic adversaries, we propose *dynamic adversary training*, a method for hardening detectors against dynamic adversaries. Based on the approach proposed by Goodfellow et al. (2015), instead of precomputing a dataset of adversarial examples, we compute the adversarial examples on-the-fly for each mini-batch and let the adversary modify each data point with probability 0.5. Note that a dynamic adversary will modify a data point differently every time it encounters the data point since it depends on the detector's gradient and the detector changes over time. We extend this approach to dynamic adversaries by employing a dynamic adversary, whose parameter $\sigma$ is selected uniform randomly from $[0, 1]$, for generating the adversarial data points during training. By training the detector in this way, we implicitly train it to resist dynamic adversaries for various values of $\sigma$. In principle, this approach bears the risk of oscillation and unlearning for $\sigma > 0$ since both, the detector and adversary, adapt to each other (i.e., there is no fixed data distribution). In practice, however, we found this approach to converge stably without requiring careful tuning of hyperparameters.

## 4 EXPERIMENTAL RESULTS

In this section, we present results on the detectability of adversarial perturbations on the CIFAR10 dataset (Krizhevsky, 2009), both for static and dynamic adversaries. Moreover, we investigate whether adversarial perturbations are also detectable in higher-resolution images based on a subset of the ImageNet dataset (Russakovsky et al., 2015).

### 4.1 CIFAR10

We use a 32-layer Residual Network (He et al., 2016, ResNet) as classifier. The structure of the network is shown in Figure 1. The network has been trained for 100 epochs with stochastic gradient descent and momentum on 45000 data points from the train set. The momentum term was set to 0.9 and the initial learning rate was set to 0.1, reduced to 0.01 after 41 epochs, and further reduced to 0.001 after 61 epochs. After each epoch, the network's performance on the validation data (the remaining 5000 data points from the train set) was determined. The network with maximal performance on the validation data was used in the subsequent experiments (with all tunable weights being fixed). This network's accuracy on non-adversarial test data is 91.3%. We attach an adversary detection subnetwork (called "detector" below) to the ResNet. The detector is a convolutional neural network using batch normalization (Ioffe & Szegedy, 2015) and rectified linear units. In the experiments, we investigate different positions where the detector can be attached (see also Figure 1).

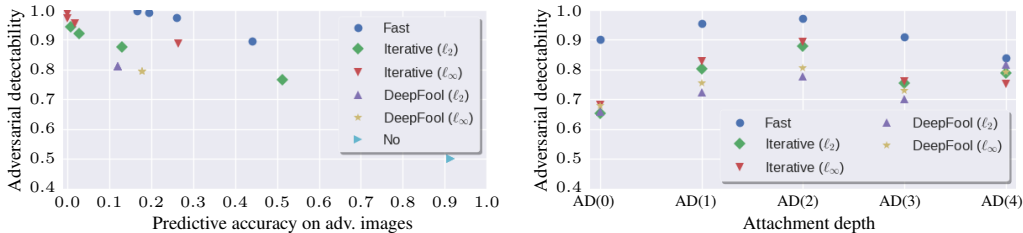

Figure 2: (Left) Illustration of detectability of different adversaries and values for $\varepsilon$ on CIFAR10. The x-axis shows the predictive accuracy of the CIFAR10 classifier on adversarial examples of the test data for different adversaries. The y-axis shows the corresponding detectability of the adversarial examples, with 0.5 corresponding to chance level. "No" corresponds to an "adversary" that leaves the input unchanged. (Right) Analysis of the detectability of adversarial examples of different adversaries for different attachment depths of the detector.

### 4.1.1 STATIC ADVERSARIES

In this subsection, we investigate a static adversary, i.e., an adversary that only has access to the classification network but not to the detector. The detector was trained for 20 epochs on 45000 data points from the train set and their corresponding adversarial examples using the Adam optimizer (Kingma & Ba, 2015) with a learning rate of 0.0001 and $\beta_1 = 0.99, \beta_2 = 0.999$. The remaining 5000 data points from the CIFAR10 train set are used as validation data and used for model selection. The detector was attached to position AD(2) (see Figure 1) except for the DeepFool-based adversaries where the detector was attached to AD(4); see below for a discussion. For the "Fast" and "Iterative" adversaries, the parameter $\varepsilon$ from Section 3.1 was chosen from $[1, 2, 3, 4]$ for $\ell_\infty$-based methods and from $[20, 40, 60, 80]$ for $\ell_2$-based methods; larger values of $\varepsilon$ generally result in reduced accuracy of the classifier but increased detectability. For the "Iterative" method with $\ell_2$-norm, we used $\alpha = 20$, i.e., in each iteration we make a step of $\ell_2$ distance 20. Please note that these values of $\varepsilon$ are based on assuming a range of $[0, 255]$ per color channel of the input.

Figure 2 (left) compares the detectability[3] of different adversaries. In general, points in the lower left of the plot correspond to stronger adversaries because their adversarial examples are harder to detect and at the same time fool the classifier on most of the images. Detecting adversarial examples works surprisingly well given that no differences are perceivable to humans for all shown settings: the detectability is above 80% for all adversaries which decrease classification accuracy below 30% and above 90% for adversaries which decrease classification accuracy below 10%. Comparing the different adversaries, the "Fast" adversary can generally be considered as a weak adversary, the DeepFool based methods as relatively strong adversaries, and the "Iterative" method being somewhere in-between. Moreover, the methods based on the $\ell_2$-norm are generally slightly stronger than their $\ell_\infty$-norm counter-parts.

Figure 2 (right) compares the detectability of different adversaries for detectors attached at different points to the classification network. $\varepsilon$ was chosen minimal under the constraint that the classification accuracy is below 30%. For the "Fast" and "Iterative" adversaries, the attachment position AD(2) works best, i.e., attaching to a middle layer where more abstract features are already extracted but still the full spatial resolution is maintained. For the DeepFool methods, the general pattern is similar except for AD(4), which works best for these adversaries.

Figure 3 illustrates the generalizability of trained detectors for the same adversary with different choices of $\varepsilon$: while a detector trained for large $\varepsilon$ does not generalize well to small $\varepsilon$, the other direction works reasonably well. Figure 4 shows the generalizability of detectors trained for one adversary when tested on data from other adversaries ($\varepsilon$ was chosen again minimal under the constraint that the

---

[3]Detectability refers to the accuracy of the detector. The detectability on the test data is calculated as follows: for every test sample, a corresponding adversarial example is generated. The original and the corresponding adversarial examples form a joint test set (twice the size of the original test set). This test set is shuffled and the detector is evaluated on this dataset. Original and corresponding adversarial example are thus processed independently.

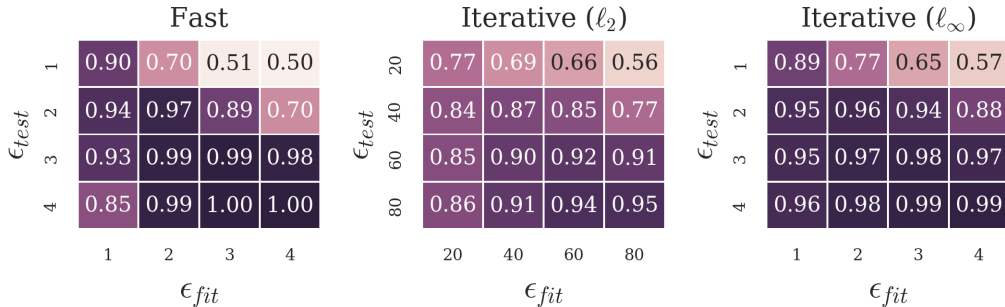

Figure 3: Transferability on CIFAR10 of detector trained for adversary with maximal distortion $\epsilon_{fit}$ when tested on the same adversary with distortion $\epsilon_{test}$. Different plots show different adversaries. Numbers correspond to the accuracy of detector on unseen test data.

| Adversary test | Fast | Iterative ($\ell_\infty$) | Iterative ($\ell_2$) | DeepFool ($\ell_2$) | DeepFool ($\ell_\infty$) |
|---|---|---|---|---|---|
| Fast | 0.97 | 0.96 | 0.92 | 0.71 | 0.75 |
| Iterative ($\ell_\infty$) | 0.69 | 0.89 | 0.87 | 0.65 | 0.68 |
| Iterative ($\ell_2$) | 0.61 | 0.79 | 0.87 | 0.59 | 0.63 |
| DeepFool ($\ell_2$) | 0.61 | 0.69 | 0.76 | 0.82 | 0.80 |
| DeepFool ($\ell_\infty$) | 0.68 | 0.80 | 0.80 | 0.78 | 0.79 |

Adversary fit

Figure 4: Transferability on CIFAR10 of detector trained for one adversary when tested on other adversaries. The maximal distortion $\epsilon$ of the adversary (when applicable) has been chosen minimally such that the predictive accuracy of the classifier is below 30%. Numbers correspond to the accuracy of the detector on unseen test data.

classification accuracy is below 30%): we can see that detectors generalize well between $\ell_\infty$- and $\ell_2$-norm based variants of the same approach. Moreover, detectors trained on the stronger "Iterative" adversary generalize well to the weaker "Fast" adversary but not vice versa. Detectors trained for the DeepFool-based methods do not generalize well to other adversaries; however, detectors trained for the "Iterative" adversaries generalize relatively well to the DeepFool adversaries.

### 4.1.2 DYNAMIC ADVERSARIES

In this section, we evaluate the robustness of detector networks to dynamic adversaries (see Section 3.3). For this, we evaluate the detectability of dynamic adversaries for $\sigma \in \{0.0, 0.1, \ldots, 1.0\}$. We use the same optimizer and detector network as in Section 4.1.1. When evaluating the detectability of dynamic adversaries with $\sigma$ close to 1, we need to take into account that the adversary might choose to solely focus on fooling the detector, which is trivially achieved by leaving the input unmodified. Thus, we ignore adversarial examples that do not cause a misclassification in the evaluation of the detector and evaluate the detector's accuracy on regular data versus the successful adversarial examples. Figure 5 shows the results of a dynamic adversary with $\varepsilon = 1$ against a static detector, which was trained to only detect static adversaries, and a dynamic detector, which was explicitly trained to resist dynamic adversaries. As can be seen, the static detector is not robust to dynamic adversaries since for certain values of $\sigma$, namely $\sigma = 0.3$ and $\sigma = 0.4$, the detectability is close to

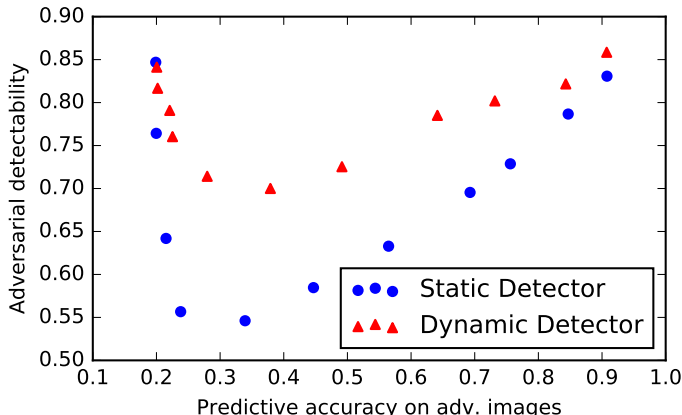

Figure 5: Illustration of detectability versus classification accuracy of a dynamic adversary for different values of $\sigma$ against a static and dynamic detector. The parameter $\sigma$ has been chosen as $\sigma \in \{0.0, 0.1, \ldots, 1.0\}$, with smaller values of $\sigma$ corresponding to lower predictive accuracy, i.e., being further on the left.

chance level while the predictive performance of the classifier is severely reduced to less than 30% accuracy. A dynamic detector is considerably more robust and achieves a detectability of more than 70% for any choice of $\sigma$.

## 4.2 10-CLASS IMAGENET

In this section, we report results for static adversaries on a subset of ImageNet consisting of all data from ten randomly selected classes[4]. The motivation for this section is to investigate whether adversarial perturbations can be detected in higher-resolution images and for other network architectures than residual networks. We limit the experiment to ten classes in order to keep the computational resources required for computing the adversarial examples small and avoid having too similar classes which would oversimplify the task for the adversary. We use a pretrained VGG16 (Simonyan & Zisserman, 2015) as classification network and add a layer before the softmax which selects only the 10 relevant class entries from the logits vector. Based on preliminary experiments, we attach the detector network after the fourth max-pooling layer. The detector network consists of a sequence of five 3x3 convolutions with 196 feature maps each using batch-normalization and rectified linear units, followed by a 1x1 convolution which maps onto the 10 classes, global-average pooling, and a softmax layer. An additional 2x2 max-pooling layer is added after the first convolution. Note that we did not tune the specific details of the detector network; other topologies might perform better than the results reported below. When applicable, we vary $\varepsilon \in [2, 4, 6]$ for $\ell_\infty$-based methods and $\varepsilon \in [400, 800, 1200]$ for $\ell_2$. Moreover, we limit changes of the DeepFool adversaries to an $\ell_\infty$ distance of 6 since the adversary would otherwise sometimes generate distortions which are clearly perceptible. We train the detector for 500 epochs using the Adam optimizer with a learning rate of $0.0001$ and $\beta_1 = 0.99, \beta_2 = 0.999$.

Figure 6 compares the detectability of different static adversaries. All adversaries fail to decrease predictive accuracy of the classifier below the chance level of $0.1$ (note that predictive accuracy refers to the accuracy on the 10-class problem not on the full 1000-class problem) for the given values of $\varepsilon$. Nevertheless, detectability is 85% percent or more with the exception of the "Iterative" $\ell_2$-based adversary with $\varepsilon = 400$. For this adversary, the detector only reaches chance level. Other choices of the detector's attachment depth, internal structure, or hyperparameters of the optimizer might achieve

---

[4]The synsets of the selected classes are: palace; joystick; bee; dugong, Dugong dugon; cardigan; modem; confectionery, confectionary, candy store; valley, vale; Persian cat; stone wall. Classes were selected by randomly drawing 10 ILSVRC2012 Synset-IDs (i.e. integers from $[1, 1000]$), using the *randint* function of the python-package *numpy* after initializing numpy's random number generator seed with 0. This results in a train set of 10000 images, a validation set of 2848 images, and a test set (from ImageNet's validation data) of 500 images.

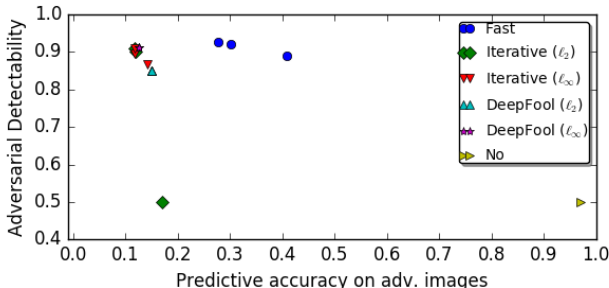

Figure 6: Illustration of detectability of different adversaries and values for $\varepsilon$ on 10-class ImageNet. The x-axis shows the predictive accuracy of the ImageNet classifier on adversarial examples of the test data for different adversaries. The y-axis shows the corresponding detectability of the adversarial examples, with 0.5 corresponding to chance level.

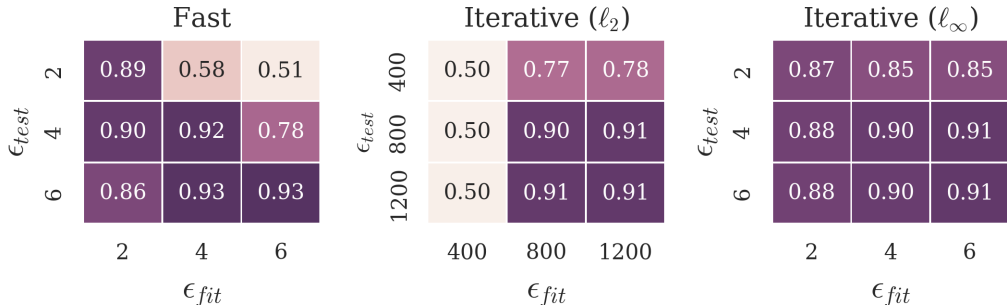

Figure 7: Transferability on 10-class ImageNet of detector trained for adversary with maximal distortion $\epsilon_{fit}$ when tested on the same adversary with distortion $\epsilon_{test}$. Different plots show different adversaries. Numbers correspond to the accuracy of the detector on unseen test data.

better results; however, this failure case emphasizes that the detector has to detect very subtle patterns and the optimizer might get stuck in bad local optima or plateaus.

Figure 7 illustrates the transferability of the detector between different values of $\varepsilon$. The results are roughly analogous to the results on CIFAR10 in Section 4.1.1: detectors trained for an adversary for a small value of $\varepsilon$ work well for the same adversary with larger $\varepsilon$ but not vice versa. Note that a detector trained for the "Iterative" $\ell_2$-based adversary with $\varepsilon = 1200$ can detect the changes of the same adversary with $\varepsilon = 400$ with 78% accuracy; this emphasizes that this adversary is not principally undetectable but that rather the optimization of a detector for this setting is difficult. Figure 8 shows the transferability between adversaries: transferring the detector works well between similar adversaries such as between the two DeepFool adversaries and between the Fast and Iterative adversary based on the $\ell_\infty$ distance. Moreover, detectors trained for DeepFool adversaries work well on all other adversaries. In summary, transferability is not symmetric and typically works best between similar adversaries and from stronger to weaker adversary.

## 5 DISCUSSION

Why can tiny adversarial perturbations be detected that well? Adopting the boundary tilting perspective of Tanay & Griffin (2016), strong adversarial examples occur in situations in which classification boundaries are tilted against the data manifold such that they lie close and nearly parallel to the data manifold. A detector could (potentially) identify adversarial examples by detecting inputs which are slightly off the data manifold's center in the direction of a nearby class boundary. Thus, the detector can focus on detecting inputs which move away from the data manifold *in a certain direction*, namely one of the directions to a nearby class boundary (the detector does not have explicit

| Adversary test \ Adversary fit | Fast | Iterative ($\ell_\infty$) | Iterative ($\ell_2$) | DeepFool ($\ell_2$) | DeepFool ($\ell_\infty$) |
|---|---|---|---|---|---|
| Fast | 0.89 | 0.88 | 0.63 | 0.84 | 0.89 |
| Iterative ($\ell_\infty$) | 0.84 | 0.87 | 0.61 | 0.81 | 0.89 |
| Iterative ($\ell_2$) | 0.66 | 0.74 | 0.90 | 0.88 | 0.87 |
| DeepFool ($\ell_2$) | 0.61 | 0.66 | 0.78 | 0.85 | 0.81 |
| DeepFool ($\ell_\infty$) | 0.80 | 0.83 | 0.69 | 0.83 | 0.91 |

Figure 8: Transferability on 10-class ImageNet of detector trained for one adversary when tested on other adversaries. The maximal distortion of the $\ell_\infty$-based Iterative adversary has been chosen as $\varepsilon = 2$ and as $\varepsilon = 800$ for the $\ell_2$-based adversary. Numbers correspond to the accuracy of detector on unseen test data.

knowledge of class boundaries but it might learn about their direction implicitly from the adversarial training data). However, training a detector which captures these directions in a model with small capacity and generalizes to unseen data requires certain regularities in adversarial perturbations. The results of Moosavi-Dezfooli et al. (2016a) suggest that there may exist regularities in the adversarial perturbations since universal perturbations exist. However, these perturbations are not unique and data-dependent adversaries might potentially choose among many different possible perturbations in a non-regular way, which would be hard to detect. Our positive results on detectability suggest that this is not the case for the tested adversaries. Thus, our results are somewhat complementary to Moosavi-Dezfooli et al. (2016a): while they show that universal, image-agnostic perturbations exist, we show that image-dependent perturbations are sufficiently regular to be detectable. Whether a detector generalizes over different adversaries depends mainly on whether the adversaries choose among many different possible perturbations in a consistent way.

Why is the joint classifier/detector system harder to fool? For a static detector, there might be areas which are adversarial to both classifier and detector; however, this will be a (small) subset of the areas which are adversarial to the classifier alone. Nevertheless, results in Section 4.1.2 show that such a static detector can be fooled along with the classifier. However, a dynamic detector is considerably harder to fool: on the one hand, it might further reduce the number of areas which are both adversarial to classifier and detector. On the other hand, the areas which are adversarial to the detector might become increasingly non-regular and difficult to find by gradient descent-based adversaries.

## 6 CONCLUSION AND OUTLOOK

In this paper, we have shown empirically that adversarial examples can be detected surprisingly well using a detector subnetwork attached to the main classification network. While this does not directly allow classifying adversarial examples correctly, it allows mitigating adversarial attacks against machine learning systems by resorting to fallback solutions, e.g., a face recognition might request human intervention when verifying a person's identity and detecting a potential adversarial attack. Moreover, being able to detect adversarial perturbations may in the future enable a better understanding of adversarial examples by applying network introspection to the detector network. Furthermore, the gradient propagated back through the detector may be used as a source of regularization of the classifier against adversarial examples. We leave this to future work. Additional future work will be developing stronger adversaries that are harder to detect by adding effective randomization which would make selection of adversarial perturbations less regular. Finally, developing methods for training detectors explicitly such that they can detect many different kinds of attacks reliably at the same time would be essential for safety- and security-related applications.

ACKNOWLEDGMENTS

We would like to thank Michael Herman and Michael Pfeiffer for helpful discussions and their feedback on drafts of this article. Moreover, we would like to thank the developers of Theano (The Theano Development Team, 2016), keras (`https://keras.io`), and seaborn (`http://seaborn.pydata.org/`).

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
