# Peer review of "On Detecting Adversarial Perturbations"

_ICLR 2017 — accepted_

[Official Review · AnonReviewer2 · rating 5 · confidence 3 · 13 Dec 2016]
**Official review**

I reviewed the manuscript on December 5th.

Summary:
The authors investigate the phenomenon of adversarial perturbations and ask whether one may build a system to independently detect an adversarial data point -- if one could detect an adversarial example, then might prevent a machine from automatically processing it. Importantly, the authors investigate whether it is possible to build an adversarial detector which is resilient to adversarial examples built against *both* the classifier and the detector. Their results suggest that training a detector in this more difficult setting still yields gains but does not entirely resolve the problem of detecting adversarial examples.

Major comments:

The authors describe a novel approach for dealing with adversarial examples from a security standpoint -- namely, build an independent system to detect the adversary so a human might intervene in those cases. 

A potential confound of this approach is that an adversary might respond by constructing adversarial examples to fool *both* the original classifier and the new detector. If that were possible, then this approach is moot since an attacker could always outwit the original system. To their credit, the authors show that building a 'dynamic' detector to detect adversarial examples but also be resilient to an adversary mitigates this potential escalation (worse case from 55% to 70% detection rate). Even though the 'dynamic' detector  demonstrates positive gains, I am concerned about overall scores. Detecting adversarial examples at this rate would not be a reliable security procedure.

My second comment is about 'model transferability'. My definition of 'model transferability' is different then the one used in the paper. My definition means that one constructs an adversarial example on one network and measures how well the adversarial examples attack a second trained model -- where the second model has been trained with different initial conditions. (The author's definition of 'transferability' is based on seeing how well the detector generalizes across training methods). 'Model transferability' (per my definition) is quite important because it measures how general an adversarial example is across all models -- and not specific to a given trained model. Different methods have different levels of 'model transferability' (Kurakin et al, 2016) and I am concerned how well the detector they built would be able to detect adversarial examples across *all models* and not just the trained model in question. In other words, a good detector would be able to detect adversarial examples from any network and not just one particularly trained network. This question seems largely unaddressed in this paper but perhaps I missed some subtle point in their descriptions.

Minor comments:

If there were any points in the bottom-left of the Figure 2 left, then this would be very important to see -- perhaps move the legend to highlight if the area contains no points.

- X-axis label is wrong in Figure 2 right.

Measure the transferability of the detector?

- How is \sigma labeled on Figure 5?

- Whenever an image is constructed to be an 'adversary', has the image actually been tested to see if it is adversarial? In other words, does the adversarial image actually result in a misclassification by the original network?

[Official Review · AnonReviewer3 · rating 7 · confidence 4 · 15 Dec 2016 (modified: 16 Dec 2016)]
**Nicely written experimental paper, making the next step in the adversarial contest.**

This paper explores an important angle to adversarial examples: the detection of adversarial images and their utilization for trainig more robust networks.

This takes the competition between adversaries and models to a new level. The paper presents appealing evidence for the feasibility of robustifying networks by employing the a detector subnetwork that is trained particularly for the purpose of detecting the adversaries in a terget manner rather than just making the networks themselves robust to adversarial examples.

The jointly trained primary/detector system is evaluated in various scenarios including the cases when the adversary generator has access to the model and those where they are generated in a generic way.

The results of the paper show good improvements with the approach and present well motived thorough analyses to back the main message. The writing is clear and concise.

[Official Review · AnonReviewer1 · rating 7 · confidence 4 · 17 Dec 2016]
**Good paper with significant novelty**

This paper proposes a new idea to help defending adversarial examples by training a complementary classifier to detect them. The results of the paper show that adversarial examples in fact can be easily detected. Moreover, such detector generalizes well to other similar or weaker adversarial examples. The idea of this paper is simple but non-trivial. While no final scheme is proposed in the paper how this idea can help in building defensive systems, it actually provides a potential new direction. Based on its novelty, I suggest an acceptance.

My main concern of this paper is about its completeness. No effective method is reported in the paper to defend the dynamic adversaries. It could be difficult to do so, but rather the paper doesn’t seem to put much effort to investigate this part. How difficult it is to defend the dynamic adversaries is an important and interesting question following the conclusions of this paper. Such investigation may essentially help improve our understanding of adversarial examples.
That being said, the novelty of this paper is still significant.

Minor comment:
The paper needs to improve its clarity. Some important details are skipped in the paper. For example, the paper should provide more details about the dynamic adversaries and the dynamic adversary training method.

[Author Response · Jan Hendrik Metzen · 10 Jan 2017]
**New revision of paper**

We have uploaded a new revision of the paper in which we have tried to address the reviewer comments. Here is a more detailed changelog:

* Fixed a bug in the ImageNet experiment: we originally applied the softmax operator twice (once before and once after selecting the ten target classes). This did not affect the accuracy of the classification network but made the network harder to fool by adversaries for similar reasons as in the ``defensive distillation'' approach. We have corrected the issue in the updated version of the paper by applying softmax only after selection the ten target classes. To briefly summarize the corrected results: adversaries remain detectable with an accuracy of at least 85% (with the same exception as before, the basic iterative l2-based adversary for epsilon=400). More details are contained in the updated Section 4.2. Sorry for this error in the first revision.
* Fixed wrong resolution in Figure 1 (16x16 instead of 8x8). Thanks to AnonReviewer3 for noting this.
* Input range specified to be [0, 255] (Section 4.1.1). Thanks to AnonReviewer1 for requesting clarification on this.
* Clarified computation of adversarial detectability (footnote in Section 4.1.1).
* We discuss briefly that dynamic adversaries are based on stronger assumptions than static adversaries (footnote in Section 3.3)
* Clarified that we did use version 1 of DeepFool (Section 3.1)
* Fixed x-axis label in Figure 2 (right). Thanks to AnonReviewer2 for noting this.
* Moved legend in Figure 2 (left) to upper right corner based on suggestion of AnonReviewer2.
* Clarified choices of \sigma in Figure 5
* Adding more details about the dynamic adversary training method.

[Final Decision · Program Chairs · 06 Feb 2017]
**ICLR committee final decision**

The paper explores the automatic detection of adversarial examples by training a classifier to recognize them. This is an interesting direction, even though they are obviously concerns about training an adversary to circumvent this model. Nonetheless, the experimental results presented in the paper are of interest to the ICLR audience. Many of the initial reviewer comments appear to be appropriately addressed in the revision of the paper.